# Explicit Alignment and Many-to-many Entailment Based Reasoning for Conversational Machine Reading

**Yangyang Luo, Shiyu Tian, Caixia Yuan, Xiaojie Wang**[*]
School of Artificial Intelligence, Beijing University of Posts and Telecommunications
{luoyangyang, tiansy, yuancx, xjwang}@bupt.edu.cn

## Abstract

Conversational Machine Reading (CMR) requires answering a user's initial question through multi-turn dialogue interactions based on a given document. Although there exist many effective methods, they largely neglected the alignment between the *document* and the *user-provided information*, which significantly affects the intermediate decision-making and subsequent follow-up question generation. To address this issue, we propose a pipeline framework that (1) aligns the aforementioned two sides in an explicit way, (2) makes decisions using a lightweight many-to-many entailment reasoning module, and (3) directly generates follow-up questions based on the document and previously asked questions. Our proposed method achieves state-of-the-art in micro-accuracy and ranks the first place on the public leaderboard[1] of the CMR benchmark dataset ShARC.

## 1 Introduction

The Conversational Machine Reading (CMR) task (Saeidi et al., 2018) requires an agent to answer an initial question from users through multi-turn dialogue interactions based on a given document. As shown in Figure 1, a typical process involves two steps, (1) the agent first makes a decision classification among *IRRELEVANT*, *YES*, *NO* and *MORE*, (2) if the decision is *MORE*, the agent generates a question to clarify an unmentioned condition in the given document, otherwise responds directly. Recent research (Verma et al., 2020; Lawrence et al., 2019; Zhong and Zettlemoyer, 2019; Gao et al., 2020a; Gao et al., 2020b; Ouyang et al., 2021; Zhang et al., 2022) has explored how to improve the abilities of *decision-making* and *question generation*.

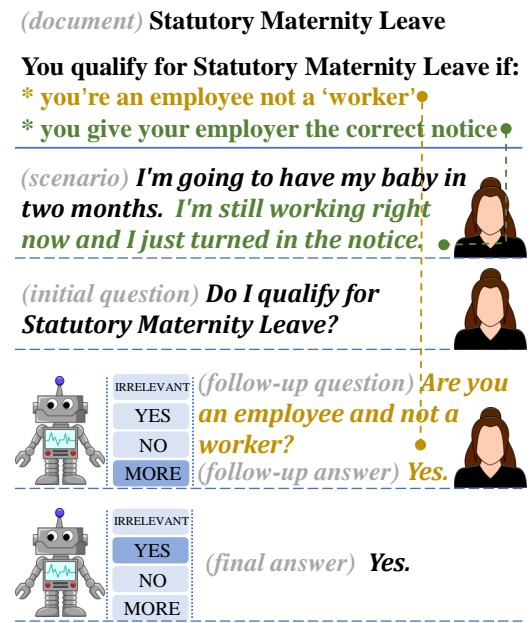

Figure 1: An example of Conversational Machine Reading from ShARC dataset (Saeidi et al., 2018).

For *decision-making*, one common approach first segments the document into many text spans at different granularity levels (e.g., sentences or Elementary Discourse Units (EDUs)). Then complex modules are adopted to predict the entailment state for each document span based on user scenario and previous dialogue history (both are user-provided information). Finally, decisions are made based on the entailment states of all document spans. One effective module for predicting entailment states is transformer blocks (Vaswani et al., 2017), which are widely adopted (Gao et al., 2020b; Ouyang et al., 2021; Zhang et al., 2022). However, the aforementioned approach has overlooked the explicit alignment between the document and the user-provided information, such as text spans marked with the same color as shown in Figure 1. Since not all user-provided information is relevant to a particular document span, the lack of explicit alignment leads to sparse attention and introduces noises that

---

*Corresponding author

[1]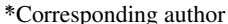https://sharc-data.github.io/leaderboard.html

affect the prediction of the entailment state. Furthermore, recent work (Ouyang et al., 2021) tries to leverage relational graph convolutional networks, which results in a heavyweight decision module, therefore greatly imposing a substantial burden on computation and memory resources.

For *question generation*, most works (Zhong and Zettlemoyer, 2019; Gao et al., 2020a; Gao et al., 2020b; Ouyang et al., 2021) first extract an unmentioned span in the document and then rewrite it into a follow-up question. The extract-then-rewrite method relies heavily on extracted spans, the failure to properly extract an unmentioned span results in generating a redundant or even irrelevant follow-up question.

To address these issues, we propose a pipeline approach consisting of a reasoning model based on **Bi**partite **A**lignment and many-to-many **E**ntailment (BiAE) for *decision-making*, and a directly fine-tuned model for *question generation* [2]. Our approach (1) explicitly aligns the document and the user-provided information by introducing supervison from an external model, (2) uses a lightweight core decision module with only linear layers, which predicts many-to-many entailment states using aligned information and feature vectors, (3) directly uses the whole document and previously asked questions to generate follow-up questions without extracting underspecified document spans. Through extensive experiments on the CMR benchmark dataset ShARC (Saeidi et al., 2018), we demonstrate that BiAE significantly outperforms baselines with lightweight decision modules by at least 12.7% in micro accuracy and the finetuned model outperforms all baselines using extract-then-rewrite generation method. Our contributions can be summarized as follows:

- We propose a method for constructing bipartite connection, which provides explicit alignment for document and user provided information.

- We propose the BiAE model which utilizes a lightweight module to make decisions by introducing explicit alignment, and a direct method which generates questions using the document and previously asked questions.

- Our approach ranks the first place on the leaderboard of ShARC and outperforms the

---

[2] https://github.com/AidenYo/BiAE

previous SOTA method on key metrics, with a significant reduction in decision module parameters.

## 2 Related Work

Machine Reading Comprehension (MRC) is a classical and fruitful research field with various tasks and focuses, such as extractive tasks (Rajpurkar et al., 2016; Trischler et al., 2017; Joshi et al., 2017; Saha et al., 2018), cloze-style and multiple choice tasks (Xie et al., 2018; Hermann et al., 2015; Richardson et al., 2013; Lai et al., 2017; Onishi et al., 2016), multi-document tasks (Feng et al., 2021; Nguyen et al., 2016; Qiu et al., 2022; Dhingra et al., 2017). Among them, we focus on Conversational Machine Reading (CMR) (Saeidi et al., 2018), which is a critical but more challenging task: (1) it requires determining complex intermediate states, such as whether the document is relevant to user's query, or whether it is necessary to make clarification before answering; (2) it requires multiple interactions with the user through dialogue in order to output final answers; and (3) the document that the agent has to consult about usually has complicated discourse structures describing multiple rules and constraints.

Due to the characteristic of determining the state before responding, the pipeline method consisting of *decision-making* and *question generation* is more suitable for this task, which is adopted by most existing methods and achieves great success (Zhong and Zettlemoyer, 2019; Gao et al., 2020a; Gao et al., 2020b; Ouyang et al., 2021). In order to improve the ability of *decision-making*, Ouyang et al. (2021) focus on improving the representation of the document and use relational GCNs (Schlichtkrull et al., 2018) to construct the discourse relations of the document. Other works focus on reasoning the entailment state of document rules, which is highly relevant to Recognizing Textual Entailment (RTE) (Bowman et al., 2015; Mou et al., 2016; Zhang et al., 2020; Wang et al., 2021). To do this, Gao et al. (2020a) modify a Recurrent Entity Network (Henaff et al., 2017), Gao et al. (2020b) use a Transformer encoder, and Zhang et al. (2022) use a T5 decoder.

To improve the ability of *question generation*, existing works (Zhong and Zettlemoyer, 2019; Gao et al., 2020a; Ouyang et al., 2021) extract a span and then rewrite it into a follow-up question, which heavily relies on the quality of the extraction.

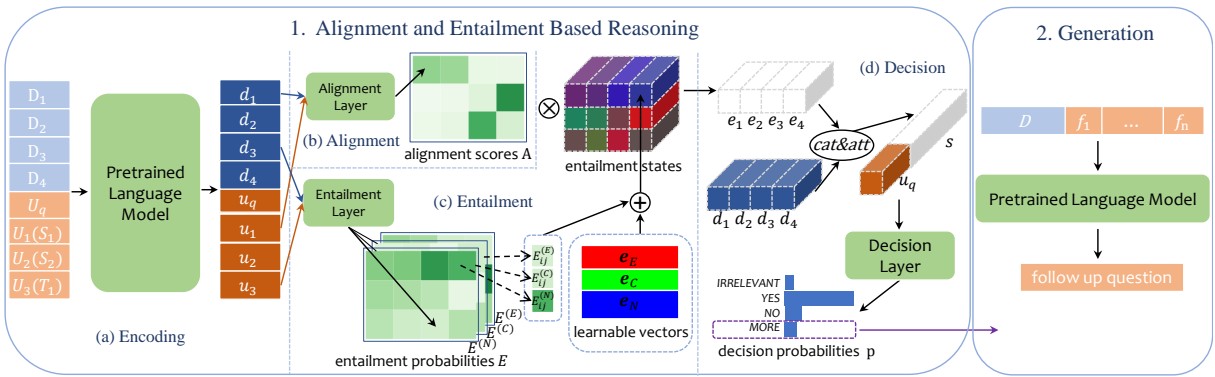

Figure 2: The overall architecture of our proposed method. Our system consists of: (1) a decision classification model based on Bipartite Alignment and Entailment (BiAE) and (2) a follow-up question generation model. It is noteworthy that only when the classification result of BiAE is *MORE* will the question generation model be activated. $cat\&att$ in 1(d) is the operation of concatenation and attention.

In comparison with these works, our work focuses on the explicit alignment of information from both the document and the user, and employs a simpler entailment reasoning structure. Then we adopt a new approach to directly generate follow-up questions based on the document and the questions asked.

## 3 Methodology

The CMR task can be formulated as follows: give input $X = (D, Q, S, H)$, $D$ is the document, $Q$ is the user's initial question, $S$ is the user's scenario, $H = (f_1, a_1), \cdots, (f_{n_H}, a_{n_H})$, $f_i$ is a follow-up question that was already asked, $a_i \in \{YES, NO\}$, is the dialogue history, a CMR system $G$ makes a response $Y = G(X)$.

We propose a classification model based on Bipartite Alignment and Entailment (BiAE) for intermediate *decision-making*. If the decision is *IRRELEVANT*, *YES* or *NO*, the system provides a direct response. If the decision is *MORE*, a fine-tuned model is used for *question generation*. The overall architecture of classification and generation is displayed in Figure 2.

### 3.1 Segmentation and Encoding

Assuming that the document is a set of hypotheses and the user-provided information is a set of premises, a segmentation step is taken first to construct hypothesis and premise sets before encoding. We not only segment documents (Gao et al., 2020b), but also make clear segmentation of user-provided information. Figure 3 shows an example of how both parts in Figure 1 is segmented.
**Segmentation.** Following Gao et al. (2020b), we

use the Segbot (Li et al., 2018) to divide the document into several Elementary Discourse Units (EDUs), with each EDU containing exactly one condition. Suppose a document can be divided into $m$ EDUs and these EDUs constitute a hypothesis set $\mathbf{D}$. $D \to \mathbf{D} : \mathbf{D_1}, \cdots, \mathbf{D_m}$.

We divide the scenario into individual sentences using NLTK[3]. $S \to \mathbf{S} : \mathbf{S_1}, \mathbf{S_2}, \cdots$. We concatenate the follow-up question and answer in a dialogue turn and add the roles in the conversation to form a premise provided by the user. $H \to \mathbf{T} : \mathbf{T_1}, \mathbf{T_2}, \cdots$, where $\mathbf{T_i} = $ *"System:"* $f_i$ *"Client:"* $a_i$. The two parts combined form the premise set $\mathbf{U} = \mathbf{S}; \mathbf{T}$ with a total number of $n$.

**Encoding.** As shown in Figure 2.1(a), we use a pre-trained language model (PLM) to encode the hypothesis set $\mathbf{D}$, initial question $U_q$, and premise set $\mathbf{U}$. We insert a special token $[H]$ before each hypothesis $\mathbf{D_i}$, and $[CLS]$ before both the initial question and each premise $\mathbf{U_i}$, separate the two parts by $[SEP]$, resulting in the input of PLM as $X$ with the length of $L$. The encoding of the input sequence is

$$encoding = PLM(X) \in \mathbb{R}^{L \times d} \qquad (1)$$

where $d$ is the dimension of PLM hidden state. The representation of each hypothesis $\mathbf{d_i}$, initial question $\mathbf{u_q}$ and premise $\mathbf{u_i}$ is determined by selecting the vector of special tokens $[H]$ and $[CLS]$ in $encoding$. More specifically,

$$\mathbf{d_i} = Select(encoding, Index(\mathbf{D_i})) \in \mathbb{R}^d, \quad (2)$$

$$\mathbf{u_i} = Select(encoding, Index(\mathbf{U_i})) \in \mathbb{R}^d, \quad (3)$$

---

[3] https://www.nltk.org

Figure 3: An example of document, scenario and conversation history segmentation.

where $Select(encoding, i)$ denotes selecting the hidden state at index $i$ from $encoding$, and $Index(\cdot)$ denotes the index of $\cdot$ in the input sequence $X$. We use DeBERTaV3 (He et al., 2021) as the PLM.

## 3.2 Explicit Alignment

The objective of explicit alignment is to align a document hypothesis $\mathbf{d_i}$ that describes a certain condition to a premise $\mathbf{u_j}$ provided by the user. We calculate the unnormalized alignment matrix $\hat{A}$ for each hypothesis-premise pair $(\mathbf{d_i}, \mathbf{u_j})$ by the following formula:

$$\hat{A}_{ij} = \mathbf{w_A}[\mathbf{d_i}; \mathbf{u_j}]^T + \mathbf{b_A} \in \mathbb{R}, \quad (4)$$

where $\mathbf{w_A}$ and $\mathbf{b_A}$ are parameters of a linear layer, $\hat{A}_{ij}$ is the hidden value of each element in the alignment matrix. Then we use the softmax function for each row to get the final alignment score matrix $A$ as shown in Figure 2.1(b),

$$A_i = softmax(\hat{A}_i) \in \mathbb{R}^n, \quad (5)$$

where the element $A_{ij} \in [0, 1]$. We use contrastive learning to train bipartite alignment and the loss can be formulated as

$$\mathcal{L}_{align} = \sum_{i=1}^{m} H(A_i, l_i^{align}), \quad (6)$$

where $H(p, q)$ represents the cross-entropy function, $l_i^{align}$ is the weakly supervised alignment label.

In order to construct $l_i^{align}$, we use Sentence-BERT (Reimers and Gurevych, 2019) to compute the semantic similarity between the user provided premise set and the document hypothesis set offline. Specifically, we calculate the cosine distance

between sentence vector pairs and select the hypothesis with the maximal cosine distance as the alignment label for each user premise.[4]

## 3.3 Many-to-many Entailment

The textual entailment task involves inferring the relationship of hypothesis-premise pair, which is generally classified into three categories: entailment, contradiction, and neutral (MacCartney and Manning, 2008). Entailment refers to the case where the hypothesis can be inferred from the premise, taking Figure 1 as an example, the user's premise *"I'm still working right now and I just turned in the notice."* entails the document hypothesis *"(You qualify for Statutory Maternity Leave if) you give your employer the correct notice"*. Contradiction represents the case where the hypothesis contradicts the premise, while neutral indicates that the relationship of the hypothesis-premise pair is unknown or irrelevant. Inspired by Mou et al. (2016), we adopt four simple yet effective features to predict the entailment states as shown in Figure 2.1(c). Specifically, we initialize three learnable vectors $\mathbf{e}_E, \mathbf{e}_C, \mathbf{e}_N \in \mathbb{R}^d$ to represent the three entailment states, use four well-designed features to predict the probabilities of the three states, and represent the entailment state of a hypothesis-premise pair as a probabilistic weighted sum of the three vectors. This process can be expressed as

$$\hat{E}_{ij} = \mathbf{W_E}[\mathbf{d_i}; \mathbf{u_j}; \mathbf{d_i} - \mathbf{u_j}; \mathbf{d_i} \circ \mathbf{u_j}]^T + \mathbf{b_E}, \quad (7)$$

$$E_{ij} = softmax(\hat{E}_{ij}) \in \mathbb{R}^3, \quad (8)$$

where $\circ$ denotes element-wise product, $\hat{E}_{ij} \in \mathbb{R}^3$ is the logits of three states, and $E_{ij} = [E_{ij}^{(E)}, E_{ij}^{(C)}, E_{ij}^{(N)}]$ is their probabilities after softmax. The final state vector for a single hypothesis across all premises weighted by alignment scores is represented as

$$\mathbf{e_i} = \sum_{j=1}^{n} A_{ij} \sum_{K \in \{E,C,N\}} E_{ij}^{(K)} \mathbf{e}_K \in \mathbb{R}^d. \quad (9)$$

The expression for the entailment loss is

$$\mathcal{L}_{entail} = \sum_{(i,j) \in \mathcal{P}} H(E_{ij}, l_{ij}^{entail}), \quad (10)$$

where $\mathcal{P}$ is the set of premise-hypothesis pairs, $l_{ij}^{entail}$ denotes the weakly supervised entailment label. We adopt the three-state label proposed by Gao et al. (2020a) to make such supervision.

---

[4]This method shows 92% consistency with manual selection on a subset of 100 randomly selected samples.

## 3.4 Decision Classification

The decision unit in Figure 2.1(d) integrates all semantic vectors and all entailment states of the hypothesis set to obtain a holistic representation $\mathbf{s}$ of the entire document, using the attention mechanism,

$$\hat{a}_i = \mathbf{w_a}[\mathbf{d_i}; \mathbf{e_i}]^T + \mathbf{b_a} \in \mathbb{R}, \quad (11)$$

$$a = softmax(\hat{a}) \in \mathbb{R}^m, a_i \in [0, 1], \quad (12)$$

$$\mathbf{s} = \sum_{i=1}^{m} a_i[\mathbf{d_i}; \mathbf{e_i}] \in \mathbb{R}^{2d}. \quad (13)$$

Subsequently, the representation $\mathbf{s}$ is employed to generate the probabilities $p$ of four aforementioned decision categories together with the semantic representation of initial question $\mathbf{u_q}$. And the corresponding decision loss is

$$p = \mathbf{W_D}[\mathbf{u_q}; \mathbf{s}]^T + \mathbf{b_D} \in \mathbb{R}^4, \quad (14)$$

$$L_{dec} = H(softmax(p), l^d), \quad (15)$$

where $l^d$ is the true decision label. Furthermore, bipartite alignment and many-to-many entailment are employed to augment the *decision-making* process, and a joint loss function is introduced incorporated with a weight parameter $\lambda$,

$$\mathcal{L} = \lambda\mathcal{L}_{dec} + \mathcal{L}_{align} + \mathcal{L}_{entail}. \quad (16)$$

## 3.5 Question Generation

If the predicted decision is *MORE*, the system is required to propose a follow-up question to obtain new premises for clarification and continuing the reasoning process. Although it is intuitive to extract a hypothesis with a neutral entailment state and then rewrite it into a clarification question, the *question generation* process heavily depends on the extracted hypothesis. Current language models, such as T5 (Raffel et al., 2020) and BART (Lewis et al., 2020), have strong generative capabilities. Hence, we directly fine-tune T5 with the entire document $D$ and the sequence of previously asked questions $F = f_1, f_2, \cdots$, while treating the ground-truth follow-up question as the generation target. We use the generation loss implemented in Raffel et al. (2020) for training.

We also perform data augmentation to alleviate data sparsity. Specifically, we reduce the dialogue history by one turn to construct $F$ for the data with decision labels other than *MORE*, and use the question in the last turn as the target question to be generated.

## 4 Experiments

## 4.1 Dataset and Metrics

**Dataset.** Our experiments are carried out on the CMR benchmark dataset ShARC (Saeidi et al., 2018), which was crawled from government legal documents across 10 unique domains. This dataset comprises 35% bullet point documents (e.g. the example shown in Figure 1), while the rest are regular documents. The dialogues are constructed based on an annotation protocol (Saeidi et al., 2018) in the form of question-answer pairs with (or not) an extra scenario. The sizes of the train, development, and test sets are 21,890, 2,270, and 8,276, respectively. The test set is withheld and not publicly available.

**Metrics.** For *decision-making*, Micro and Macro Accuracy are used for evaluation, whereas BLEU (Papineni et al., 2002) is used for evaluating *question generation*.

## 4.2 Baselines

(1) **Baseline-NMT** (Saeidi et al., 2018) is an end-to-end NMT-copy model based on LSTM and GRU. (2) **Baseline-CM** (Saeidi et al., 2018) is a pipeline combined model using Random Forest, Surface Logistic Regression and rule-based generation. (3) **BERTQA** (Zhong and Zettlemoyer, 2019) is an extractive QA model. (4) **UracNet** (Verma et al., 2020) uses artificially designed heuristic-based patterns. (5) **BiSon** (Lawrence et al., 2019) utilizes placeholders for bidirectional generation rather than autoregressive unidirectional generation. (6) $\mathbf{E}^3$ (Zhong and Zettlemoyer, 2019) performs rule extraction from documents, rule entailment from user information, and rule editing into follow-up questions jointly. (7) **EMT** (Gao et al., 2020a) uses a gated recurrent network with augmented memory that updates rule entailment state for *decision-making* by sequentially reading user information. (8) **DISCERN** (Gao et al., 2020b) subdivides a document into fine-grained EDUs and employs an inter-sentence transformer encoder for entailment prediction. (9) **DGM** (Ouyang et al., 2021) primarily employs relational GCNs (Schlichtkrull et al., 2018) to model the rhetorical structure of documents for *decision-making*. (10) **ET5** (Zhang et al., 2022) proposes an end-to-end generation approach with duplex decoders and a shared encoder based on T5.

Baselines (8), (9) and (10) use heavyweight modules for core *decision-making*. Please refer to Ap-

| Model | Held-out Test Set | | | | Dev Set($B$) | | Dev Set($L$) | |
|---|---|---|---|---|---|---|---|---|
| | *mic* | *mac* | *B-1* | *B-4* | *mic* | *mac* | *mic* | *mac* |
| NMT (Saeidi et al., 2018) | 44.8 | 42.8 | 34.0 | 7.8 | - | - | - | - |
| CM (Saeidi et al., 2018) | 61.9 | 68.9 | 54.4 | 34.4 | - | - | - | - |
| BERTQA (Zhong and Zettlemoyer, 2019) | 63.6 | 70.8 | 46.2 | 36.3 | 63.6 | 70.8 | - | - |
| UrcaNet (Verma et al., 2020) | 65.1 | 71.2 | 60.5 | 46.1 | - | - | - | - |
| BiSon (Lawrence et al., 2019) | 66.9 | 71.6 | 58.8 | 44.3 | 66.9 | 71.6 | - | - |
| E$^3$ (Zhong and Zettlemoyer, 2019) | 67.6 | 73.3 | 54.1 | 38.7 | 67.6 | 73.3 | - | - |
| EMT (Gao et al., 2020a) | 69.1 | 74.6 | 63.9 | 49.5 | 69.1 | 74.6 | - | - |
| DISCERN* (Gao et al., 2020b) | 73.2 | 78.3 | 64.0 | 49.1 | 74.9 | 79.8 | 77.2 | 80.3 |
| ET5* (Zhang et al., 2022) | 76.3 | 80.5 | 69.6$^\dagger$ | 55.2$^\dagger$ | 75.9 | 80.4 | 78.6 | 82.5 |
| DGM* (Ouyang et al., 2021) | 77.4 | **81.2** | 63.3 | 48.4 | 75.5 | 79.6 | 78.6 | 82.2 |
| BiAE(ours) | **77.9** | 81.1 | **64.7** | 51.6 | 76.2 | 80.5 | 80.5 | 83.2 |

Table 1: Results on the held-out ShARC test set and dev set. *B-1*, *B-4*, *mic* and *mac* are short for *BLEU1*, *BLEU4*, *Micro Accuracy* and *Macro Accuracy*. We conduct 5-fold cross-validation t-test experiments on the dev set since the test set is reserved, and the results show that p < 0.05. Models with * use heavyweight decision modules and $^\dagger$ ET5 uses dual decoders. The performance of each method using base and large version pre-trained language models on the development set is marked by *B* and *L*, respectively.

pendix A for implementation details.

## 5 Results and Analysis

### 5.1 Main Results

We report the results of BiAE and baselines on the blind held-out test set of the ShARC dataset in Table 1. BiAE significantly outperforms the baselines with lightweight decision modules, with at least a 12.7% improvement in Micro Accuracy and an 8.7% improvement in Macro Accuracy. Compared to baselines with heavyweight decision modules, BiAE achieves comparable results while greatly reducing the parameters from $27M$ (decision module of DGM) to only $31.7K$. Moreover, BiAE achieves state-of-the-art performance in terms of Micro Accuracy. For generation, T5$_{BiAE}$ outperforms all the methods using span extraction. Note that the generation metrics are only calculated when both the classification decision and the true label is *MORE*. Therefore, since the test sets differ among the methods used to calculate generation scores, the results are not strictly comparable.

To compare the *decision-making* ability of different methods using base and large pre-trained language models fairly, we also report the results on the development set in Table 1. Regardless of whether based on a base or large pre-trained language model, BiAE significantly outperforms the baselines with lightweight decision modules, meanwhile, it achieves higher Micro Accuracy of 1.0 (base) and 1.9 (large) and Macro Accuracy of

| Model | paras | *mic* | *mac* | *B-1* | *B-4* |
|---|---|---|---|---|---|
| FastChat-T5 | 3B | 12.7 | 29.8 | - | - |
| + 2-shot | 3B | 20.6 | 35.9 | 50.9 | 29.5 |
| Alpaca | 7B | **51.8** | 37.7 | - | - |
| + 2-shot | 7B | 41.8 | 30.2 | - | - |
| Vicuna | 13B | 24.8 | 39.0 | 11.9 | 7.7 |
| + 2-shot | 13B | 33.1 | 41.9 | 33.8 | 12.0 |
| GPT-3.5-turbo | 175B | 35.6 | 45.2 | **58.6** | **37.1** |
| + 2-shot | 175B | 41.1 | **50.8** | 45.2 | 29.5 |

Table 2: Results of large language models on the ShARC development set. *B-1*, *B-4*, *mic* and *mac* are short for *BLEU1*, *BLEU4*, *Micro Accuracy* and *Macro Accuracy*.

0.7 (base) and 1.0 (large) than the strong baseline DGM.

We also report class-wise Micro Accuracy of BiAE and several baseline models on four different categories in Appendix B. BiAE greatly improves the abilities of deterministic *decision-making* (*YES* and *NO*).

**Compare with Large Language Models.** We have evaluated the performance of FastChat-T5 (Zheng et al., 2023), Alpaca (Taori et al., 2023), Vicuna (Chiang et al., 2023) and GPT-3.5-turbo (Ouyang et al., 2022) on the ShARC development set, and the results with 0-shot and 2-shot demonstrations are reported in Table 2. The four large language models show significantly lower results compared to our model. Please refer to Appendix

| Model | micro-acc | macro-acc |
|---|---|---|
| BiAE(ELECTRA) | 76.2 | 80.3 |
| BiAE(DeBERTaV3) | 76.2 | 80.5 |
| w/o Align | 74.3 | 78.9 |
| w/o Entail | 74.1 | 78.3 |
| w/o Align & Entail | 72.6 | 76.9 |

Table 3: Results of ablation experiments for decision reasoning based on the ELECTRA-base and DeBERTaV3-base models on the ShARC development set.

| Model | BLEU1 | BLEU2 | BLEU3 | BLEU4 |
|---|---|---|---|---|
| T5$_{BiAE}$ | 62.8 | 55.8 | 51.7 | 48.6 |
| w/o aug. | 61.4 | 54.1 | 49.8 | 46.6 |

Table 4: Results of ablation experiments for follow-up generation on the ShARC development set.

C for the prompt template we used and some output cases.

## 5.2 Ablation Study

**Effect of Alignment and Entailment.** To explore the effects of bipartite alignment and many-to-many entailment on *decision-making*, we conduct ablation experiments based on DeBERTaV3-base on the development set, as shown in Table 3. The results indicate that both alignment and entailment have an impact on *decision-making* and the impact is greater when both are considered together. Simultaneously removing both alignment and entailment losses leads to a decrease of 3.6 points in Micro Accuracy and 3.4 points in Macro Accuracy. Furthermore, we also conduct ablation experiments on the encoders, the results based on ELECTRA-base exhibits a slight decrement when compared to those based on DeBERTaV3-base. This demonstrates that the improvement in reasoning performance is minimally influenced by encoders and mainly comes from our incorporation of explicit alignment and the modeling of many-to-many entailment in the core decision module.

**Effect of Data Augmentation for Generation.** Only the data with the decision label *MORE* requires generating follow-up questions, accounting for 31.08% of the training set. We perform data augmentation to address this data sparsity issue and organize ablation experiments, results are shown in Table 4. Data augmentation improves all generation metrics, with increases of 1.4, 1.7, 1.9 and 2.0 for BLEU1-4, respectively. Appendix D is a

| Subset | #Count | Subset | #Count |
|---|---|---|---|
| Bullet Point | 999 | Regular | 1271 |
| Scenario | 1839 | NoScenario | 431 |
| History | 1509 | NoHistory | 761 |
| All | 2270 | | |

Table 5: Subset Size of ShARC Development Set.

| State | $\beta$ | $\bar{\alpha}$ | $\sigma_\alpha^2$ | $Q_1^\alpha$ | $Q_2^\alpha$ | $Q_3^\alpha$ |
|---|---|---|---|---|---|---|
| **Success** | 0.45 | 0.75 | 0.29 | 0.50 | 0.80 | 1.00 |
| **Fail** | 0.24 | 0.72 | 0.24 | 0.60 | 0.75 | 0.89 |

Table 6: $\beta$ and some statistics of $\alpha$. $\sigma^2$ denotes the variance and $Q_n$ denotes the $n$-th quartile.

simple case study of generation.

## 5.3 Interpretation of Document and User Information

To investigate the comprehension ability of BiAE for document and user information, we divide the development set into six subsets based on whether the documents contain bullet points and whether the user information includes scenario or conversation history. The sizes of each subset are shown in Table 5.

We calculate the Micro and Macro Accuracy of the strong baseline DGM and BiAE on different subsets, as shown in Figure 4(a). The results show that BiAE performs better on all subsets. Understanding bullet point documents is more challenging than regular documents, but BiAE reduces the gap by improving the Micro Accuracy of bullet point documents by 3.3 and regular documents by 2.2. Understanding scenarios is still a significant challenge (Saeidi et al., 2018; Gao et al., 2020b; Ouyang et al., 2021) because subsets without scenarios have significantly higher accuracy than those with scenarios. BiAE achieves the most significant improvement (+3.9) on subsets containing history. These performance improvements are likely due to our splitting of user information and the explicit alignment between documents and user information.

## 5.4 Many-to-many Entailment

In BiAE, the final decision is based on: the encoding of user initial question, the encoding and the final entailment state of each hypothesis in a document. To investigate the holistic textual entailment of the document hypothesis set on the final

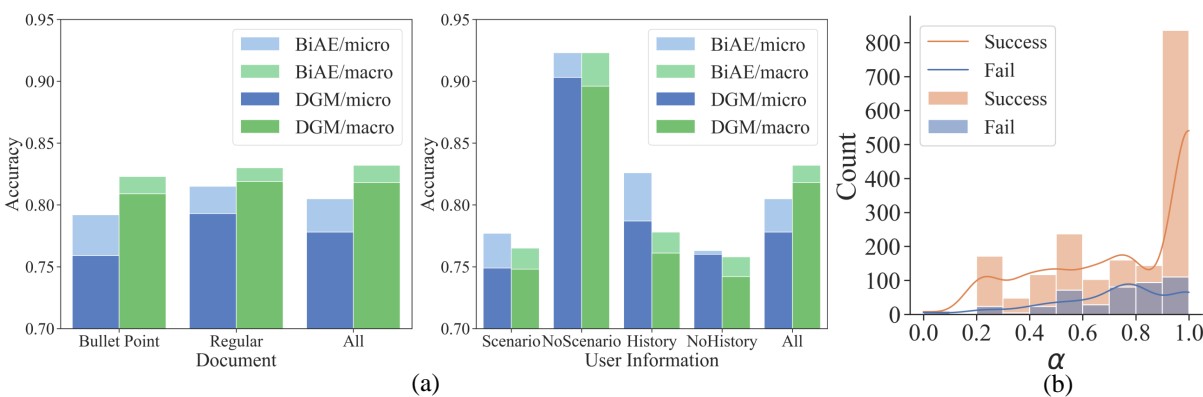

Figure 4: (a)Accuracy of BiAE and DGM(Strong Baseline) on Different Subsets. (b)Histogram and Density Estimation of the $\alpha$ Distribution under Successful and Failed Prediction States.

decision, we define $\alpha$ and $\beta$ as follows:

$$\alpha = \frac{\sum_{i=1}^{m} \mathbb{I}(s_i^p = s_i^q)}{m}, \quad (17)$$

$$\beta^{\mathcal{P}} = \frac{\sum_{\alpha \in \mathcal{P}} \mathbb{I}(\alpha = 1.0)}{|\mathcal{P}|}, \quad (18)$$

where $m$ is the number of hypotheses in a document, $\mathcal{P}$ is a subset, $s_i^p$ and $s_i^q$ is the predicted and constructed label for the $i$-th hypothesis, $\mathbb{I}(\cdot)$ denotes the indicator function. $\alpha$ measures the degree of correctness in entailment reasoning for an individual document and $\beta$ represents the proportion of documents with perfect entailment reasoning in a subset. Figure 4(b) illustrates the distribution and density estimation curve of $\alpha$ under successful and failed prediction states. The statistics in Table 6 and Figure 4(b) show that, compared with failed predictions, the values of $\alpha$ are more concentrated around 1.0 in successful predictions, indicating deeper understanding of the entailment of all hypotheses in the document (corresponding to larger $\alpha$ values), and hence leading to higher prediction accuracy. In addition, $\beta^{Sucess}$ is much larger than $\beta^{Fail}$, while the difference between $\bar{\alpha}^{Sucess}$ and $\bar{\alpha}^{Fail}$ is small, indicating that the final decision not only requires one-to-one entailment but also relies on accurate many-to-many entailment of all hypotheses in the document.

## 6  Conclusion

We propose a new framework for Conversational Machine Reading in this paper. Our classification model, BiAE, leverages many-to-many entailment reasoning, enhanced by explicit alignment, for *decision-making*. And our T5$_{\text{BiAE}}$ is directly fine-tuned for generating follow-up questions to clarify

underspecified document spans. BiAE significantly reduces the parameters in the core *decision-making* module and achieves results comparable to strong baselines. Extensive experiments demonstrate the effectiveness of our framework. Through analysis, we believe that improving the ability of entailment reasoning among the overall hypotheses of a document is crucial for enhancing *decision-making* ability.

## Limitations

Although our approach exceeds the previous state-of-the-art model on the main metrics, our work still has two limitations.

(1) We conduct experiments on the ShARC dataset, which is the benchmark for the CMR task but consists of relatively short documents. Due to limited computational resources, it is challenging for us to organize experiments with longer documents or larger datasets. However, we believe that BiAE has the potential to perform well on longer documents. We will strive to scale up our computational resources and validate our approach on such datasets.

(2) Our proposed method for explicit alignment of documents and dialogues is based on semantic similarity. Although it demonstrates effectiveness in our experiments, we acknowledge that various knowledge bases, such as knowledge graphs, can provide alignment information beyond the semantic level. In the future, we will explore better alignment methods by leveraging diverse knowledge bases.

## Ethics Statement

Our research focuses on conversational machine reading. The dataset we used is publicly available and consists of documents from government websites and dialogues from crowdsourcing workers who have received fair and legal remuneration. We utilize open-source pre-trained models, including large language models. GPT-3.5 is only used for a simple evaluation. We believe that our work does not involve personal privacy or societal biases.

## Acknowledgements

The work is partially supported by State Grid Corporation of China's Science and Technology Project "Construction of Electric Power Cognitive Large Model and key Techniques of Its Applications on Operation, Maintenance and Detection" (Project No: 5700-202313288A-1-1-ZN). We thank the anonymous reviewers for their insightful comments and Max Bartolo for executing the evaluation on the reserved test set.

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

## A    Implementation Details

Adam optimizer (Kingma and Ba, 2015) and linear schedule with warmup are used for the training process. For the *decision-making* task, BiAE is fine-tuned based on DeBERTaV3 for 5 epochs with dropout rate set to 0.3. Batch size is set to 20 for base model and 8 for large model. We try various loss weights in Equation 16, including $\lambda = 0.5, 1.0, 2.0, 3.0$, and find that 2.0 works best. The learning rates $1e^{-5}, 2e^{-5}, 5e^{-5}$, and $1e^{-4}$ are attempted, and we find that $5e^{-5}$ is optimal for base model, while $2e^{-5}$ is best for large model. For the *question generation* task, T5 large is fine-tuned for 3 epochs, with batch size set to 6, learning rate set to $2e^{-4}$, and all other parameters set to default. All experiments are conducted on a NVIDIA GeForce RTX 3090. It takes 6-7 hours to fine-tune BiAE (DeBERTaV3-large) for 5 epochs, 1-2 hours to fine-tune T5-large (without data augmentation) for 3 epochs, and 7-8 hours to fine-tune T5-large (with data augmentation) for 3 epochs. The data augmentation is performed during the construction of the Dataset and the time required for it is negligible.

## B    Deterministic and Uncertainty Reasoning

Table 7 shows Micro Accuracy of our model and several baseline models on four different categories. All models demonstrate high reasoning abilities for *IRRELEVANT* (over 95%). Compared to the strong baseline, BiAE performs slightly worse in reasoning uncertainty problems (*MORE*, the reasoning ability to clarify questions), but greatly improves the abilities of deterministic *decision-making* (*YES* and *NO*). This phenomenon may stem from the fact that the selected entailment features exhibit a higher sensitivity towards deterministic reasoning.

| Model | IRRELEVANT | YES | NO | MORE | Total |
|---|---|---|---|---|---|
| BERTQA | 96.4 | 61.2 | 61.0 | 62.6 | 63.6 |
| E$^3$ | 96.4 | 65.9 | 70.6 | 60.5 | 68.0 |
| UracNet | 95.7 | 63.3 | 68.4 | 58.9 | 65.9 |
| EMT | 98.6 | 70.5 | 73.2 | 70.8 | 74.2 |
| Discern | **99.3** | 71.9 | 75.8 | 73.3 | 75.2 |
| DGM | 97.8 | 75.2 | 77.9 | **76.3** | 77.8 |
| BiAE(ours) | 97.1 | **84.1** | **80.5** | 71.2 | **80.5** |

Table 7:  Class-wise Accuracy of BiAE and baselines on the ShARC development set.

## C    Prompt Template and Examples

We use the 0-shot prompt template as shown in Figure 5 and 2-shot prompt template as shown in Figure 8 to create inputs for large language models. Figure 6 and Figure 7 are two 0-shot output examples.

## D    Generation Case Study

We conduct a case study on 100 samples with the lowest BLEU scores to analyze the reasons, and the main categories are summarized as follows:

1. Incomplete generation of questions: 2%

2. Generated questions lacking key words: 2%

3. Generated questions lacking non-key words: 8%

4. Generated questions with the same semantics as the true questions but different expressions: 8%

5. Generated questions unrelated to the document: 13%

6. Redundant generation of questions (already asked or unnecessary): 25%

7. Generated questions describing other reasonable and unasked conditions: 42%

It should be noted that evaluating the generation task solely based on BLEU is not sufficient. Categories 4 and 7 (50% in total) represent acceptable generated questions but receive lower BLEU scores.

---

You are a customer service consultant. Give a proper response to answer the initial question proposed by the client based on the given document. You can only reply 'Irrelevant', 'Yes', 'No' directly or ask a follow-up question without explanation. Don't explain!

Document:
*[document]*

Initial question: *[scenario] [initial question]*

The following is the conversation that has already happened:
You: *[follow-up question 1]*
Client: *[follow-up answer 1]*
...

---

Figure 5: The 0-shot prompt template for large language models.

You are a customer service consultant. Give a proper response to answer the initial question proposed by the client based on the given document. You can only reply 'Irrelevant', 'Yes', 'No' directly or ask a follow-up question without explanation. Don't explain!

Document:
*If you are a female Vietnam Veteran with a child who has a birth defect or you are a child of a female Vietnam Veteran with a birth defect, the child may be eligible for VA-financed health care.*

Initial question: *I registered as a single not knowing my son would have a birth defect. Is my child eligible for VA-financed health care?*

The following is the conversation that has already happened:
You: *Are you a female Vietnam Veteran?*
Client: *Yes.*
You: *Does your child have a birth defect?*
Client: *Yes.*

**True answer:** Yes.
**Output of FastChat-T5:** Irrelevant.
**Output of Alpaca:** Yes, your child is eligible for VA-financed health care.
**Output of Vicuna:** Yes, your child may be eligible for VA-financed health care.
**Output of GPT-3.5-turbo:** Yes.

Figure 6: An output case of large language models (0-shot).

You are a customer service consultant. Give a proper response to answer the initial question proposed by the client based on the given document. You can only reply 'Irrelevant', 'Yes', 'No' directly or ask a follow-up question without explanation. Don't explain!

Document:
*## Statutory Maternity Pay*

*To qualify for SMP you must:*

- *earn on average at least £113 a week*
- *give the correct notice*
- *give proof you're pregnant*
- *have worked for your employer continuously for at least 26 weeks continuing into the 'qualifying week' - the 15th week before the expected week of childbirth'*

Initial question: *I have worked for Her Majesty's Revenue and Customs for the past three years without a single day off and am expected to have my baby in about nine weeks. Do I qualify for SMP?*

The following is the conversation that has already happened:
You: *Do you earn on average at least £113 a week?*
Client: *Yes.*
You: *Did you give the correct notice?*
Client: *Yes.*

**True answer:** Did you give proof you're pregnant?
**Output of FastChat-T5:** Irrelevant.
**Output of Alpaca:** Yes, you qualify for SMP.
**Output of Vicuna:** Based on the information provided, it appears that the client qualifies for SMP. However, it is important to note that the client must also provide proof that they are pregnant and have worked for their employer continuously for at least 26 weeks into the qualifying week.
**Output of GPT-3.5-turbo:** You: Have you provided proof that you're pregnant?

Figure 7: An output case of large language models which requires generating a follow-up question (0-shot).

You are a customer service consultant. Give a proper response to answer the initial question proposed by the client based on the given document. You can only reply 'Irrelevant', 'Yes', 'No' directly or ask a follow-up question without explanation. Don't explain!

### Example
Document:
*Eligibility*
*You must have:*

- *an unconditional offer of a place on a course with a licensed Tier 4 sponsor*
- *enough money to support yourself and pay for your course - the amount will vary depending on your circumstances.*

Initial question: *I have an unconditional offer for a course with a licensed tier 4 sponsor who is not a panel of experts. Am I eligible for a Tier 4 (General) student visa?*

The following is the already happened conversation:
You: *Do you have an unconditional offer of a place on a course with a licensed Tier 4 sponsor?*
Client: *Yes.*
You: *Do you have enough money to support yourself and pay for the course?*
Client: *No.*

The final response: ***No.***

### Example
Document:
*In order to qualify for this benefit program, your business or private non-profit organization must have sustained physical damage and be located in a disaster declared county.*

Initial question: *My housing benefit doesn't currently cover my rent. Does this program meet my needs?*

No conversation has taken place.

The final response: ***Do you own a business or private non-profit organization?***

Give a proper response for the following initial question. You can only reply 'Irrelevant', 'Yes', 'No' directly or ask a follow-up question without explanation.

Document:
*[document]*

Initial question: *[scenario] [initial question]*

The following is the conversation that has already happened:
You: *[follow-up question 1]*
Client: *[follow-up answer 1]*
...

The final response:

Figure 8: The 2-shot prompt template for large language models.