# OpenReview forum: "Explicit Alignment and Many-to-many Entailment Based Reasoning for Conversational Machine Reading"
_EMNLP/2023/Conference — EMNLP 2023 Findings_

### Official Review · Reviewer_1skF · 2023-08-05

**Soundness:** 3

**Excitement:**

2: Mediocre: This paper makes marginal contributions (vs non-contemporaneous work), so I would rather not see it in the conference.

**Paper Topic And Main Contributions:**

The paper introduces a pipeline framework for Conversational Machine Reading (CMR) that explicitly aligns the user's initial question with the provided document and utilizes a lightweight many-to-many entailment reasoning module for decision-making. The framework also enables direct follow-up question generation based on the document and previously asked questions. By addressing the alignment issue and incorporating effective reasoning and question generation techniques, the proposed method achieves state-of-the-art results on ShARC dataset.

**Reasons To Accept:**

1. The paper introduces a lightweight many-to-many entailment reasoning module for decision-making.
2. The proposed approach achieves the state-of-the-art results on ShARC dataset.

**Reasons To Reject:**

1. The novelty is limited. As mentioned in this work, most part of this paper follows the previous approach EMT, DISCERN, ET5, and DGM. The biggest contribution is the proposed lightweight many-to-many entailment reasoning module.
2. The improvement is marginal compared to previous work.

**Reproducibility:**

4: Could mostly reproduce the results, but there may be some variation because of sample variance or minor variations in their interpretation of the protocol or method.

**Reviewer Confidence:**

5: Positive that my evaluation is correct. I read the paper very carefully and I am very familiar with related work.

---

> ### Author Rebuttal · Authors · 2023-08-29
>
> Thanks for your careful and valuable comments. We will explain your concerns and answer your question point by point.
>
> # Reasons To Reject Point 1:
> The motivation behind our work can be summarized into two points. Firstly, existing methods for modeling have not taken into account the explicit alignment between document rules and user information, leading to the introduction of noise and sparse attention. Secondly, the core decision modules of existing methods (e.g., DGM using graph networks + transformers) have become increasingly complex, consuming more computational resources and memory. Therefore, in addition to the explicit alignment modeling, our main contributions include designing a many-to-many entailment analysis module, significantly reducing the parameter size of the core decision module (only 0.06% of DGM). Experimental results show that our lightweight core decision module achieves comparable results to DGM.
>
> Furthermore, we also conducted ablation experiments on the encoder of the reasoning model, as we recognize the importance of fair comparison. The results of the ablation experiments based on ELECTRA on the development set are as follows. The reasoning results based on ELECTRA are slightly lower than those based on DeBERTa but still higher than DGM. This allows for a fair comparison with the previous SOTA method, DGM, and also demonstrates that the improvement in reasoning performance mainly comes from our incorporation of explicit alignment and the modeling of many-to-many entailment in the core decision module. We will include these experimental results in the revised version. Thank you for your feedback.
> | Model               | Micro Acc. | Macro Acc. |
> |---------------------|------------|------------|
> | DGM (electra-base)  | 75.5       | 79.6       |
> | BiAE (electra-base) | 76.2       | 80.3       |
> | BiAE (DeBERTa-base) | 76.2       | 80.5       |
> | DGM (electra-large) | 78.6       | 82.2       |
> | BiAE (electra-large)| 79.9       | 82.8       |
> | BiAE (deBERTa-large)| 80.5       | 83.2       |
>
>
> # Reasons To Reject Point 2:
> In terms of model performance, our main contributions are twofold. Firstly, we significantly reduced the parameter size of the core decision module by introducing explicit alignment and many-to-many entailment, with our core decision module accounting for only 0.06% of DGM's. Secondly, we achieved comparable results to strong baselines while also surpassing the previous SOTA (DGM) on the main metric.
>
> In addition, we conducted significance test experiments. As the test set is reserved (evaluation on the test set requires submitting the model and code to the official ShARC platform), it is hard for us to perform significance tests on the test set. However, we divided the training set into five parts and conducted a 5-fold cross-validation t-test, resulting in p < 0.05. The result indicates that our performance is significantly higher than that of DGM.

---

### Official Review · Reviewer_HUMs · 2023-08-10

**Soundness:** 3

**Excitement:**

3: Ambivalent: It has merits (e.g., it reports state-of-the-art results, the idea is nice), but there are key weaknesses (e.g., it describes incremental work), and it can significantly benefit from another round of revision. However, I won't object to accepting it if my co-reviewers champion it.

**Paper Topic And Main Contributions:**

The paper proposes Bipartite Alignment and many-to-many Entailment (BiAE) for the Conversational Machine Reading (CMR) task. BiAE uses DeBERTaV3 and T5. BiAE features a lightweight decision module, showcasing a substantial performance improvement over its other small models on the ShARC benchmark. Moreover, BiAE surpasses the performance of previous state-of-the-art heavyweight models across critical metrics. The authors also highlight the importance of explicit alignment of the document and the user-provided information in CMR.

**Questions For The Authors:**

Question A: Is there any effort dedicated to checking for the quality of weakly supervised alignment labels?

Question B: The representation of segments in document D (d_1, d_2, … d_n) is often utilized independently, such as in alignment and decision processes. Is it reasonable to utilize these representational vectors independently? In the Segmentation part of Section 3 (Methodology), the Segbot does not decontextualize Elementary Discourse Units of a given document D (Refer to Decontextualization: Making Sentences Stand-Alone (Choi et al., TACL 2021) for the concept of decontextualization). Is that necessary to decontextualize these EDUs?

**Reasons To Accept:**

1.	BiAE is a novel approach to the problem of CMR. The ablation study conducted on BiAE offers valuable insights into CMR.

2.	This paper introduces a novel approach for constructing a lightweight decision module, which has significant merit for future direction of CMR.

**Reasons To Reject:**

1.	The experiments with BiAE are trivial. Experiments using only one dataset is not enough to prove the effectiveness of BiAE over baselines in this paper.

2.	It is still unclear if the superior performance by BiAE is thanks to the novel architecture proposed or the superior encoding module (DeBERTaV3). This is because previous state-of-the-art models use previous core modules, such as DGM uses ELECTRA,  ETM uses BERT base, … Further study about the core modules is needed as we know that DeBERTaV3 significantly outperform previous models on the task of RTE.

Note: If the author(s) can address any of these concerns during the rebuttal phase, the final revision may be a strong candidate for the conference.

**Reproducibility:**

3: Could reproduce the results with some difficulty. The settings of parameters are underspecified or subjectively determined; the training/evaluation data are not widely available.

**Reviewer Confidence:**

3: Pretty sure, but there's a chance I missed something. Although I have a good feel for this area in general, I did not carefully check the paper's details, e.g., the math, experimental design, or novelty.

**Typos Grammar Style And Presentation Improvements:**

In section 3, why do the author(s) have to define D = w_1 w_2…w_{n_D}, Q = w_1 w_2…w_{n_Q}, S = w_1 w_2…w_{n_S}? This paper's definition of w is unnecessary and could potentially be excluded, enhancing readers' comprehension of the fundamental concepts presented.

---

> ### Author Rebuttal · Authors · 2023-08-29
>
> Thanks for your careful and valuable comments. We will explain your concerns and answer your questions point by point.
>
> # Reasons To Reject Point 1:
> Thank you for your feedback. Due to limited computational resources, it is challenging for us to organize more experiments. Conducting experiments on additional datasets will be left for our future work. In order to provide more comprehensive evidence of the effectiveness of BiAE, we divide the training set into five parts, conducting a 5-fold cross-validation experiment. We perform a statistical significance t-test between the results of these five experiments and the results of DGM, resulting in p < 0.05. This experiment partially supplements the demonstration of the effectiveness of our approach. Thank you again for your feedback.
>
> # Reasons To Reject Point 2:
> Thank you very much for your insightful suggestion. We acknowledge that fair comparison is crucially important. Therefore, for the reasoning model, we conducted ablation experiments based on ELECTRA encoders on the development set (results shown below). The experimental results show that the reasoning results based on ELECTRA are slightly lower than those based on DeBERTa, but still higher than DGM. On one hand, this can fairly compare with the previous SOTA, DGM; on the other hand, this can illustrate that the improvement in reasoning performance mainly comes from our introduction of explicit alignment and the modeling of many-to-many entailment into the core decision module. It should be noted that the parameter size of our core decision module (31.7K) is only 0.06% of that of DGM (54M). One of our main contributions is achieving comparable results to DGM while greatly reducing the parameter size of the core decision module. We will add this ablation experiment to the revised version. Thank you again for your feedback.
> | Model               | Micro Acc. | Macro Acc. |
> |---------------------|------------|------------|
> | DGM (electra-base)  | 75.5       | 79.6       |
> | BiAE (electra-base) | 76.2       | 80.3       |
> | BiAE (DeBERTa-base) | 76.2       | 80.5       |
> | DGM (electra-large) | 78.6       | 82.2       |
> | BiAE (electra-large)| 79.9       | 82.8       |
> | BiAE (deBERTa-large)| 80.5       | 83.2       |
>
>
> # Question A:
> Thank you for your feedback. We have validated the quality of alignment labels through two methods. Firstly, we use a heuristic approach to make labels. Specifically, we calculate the edit distance between all document rules and user information and assign the document rule with the minimum edit distance as the label. The labels generated by the heuristic method show 85% consistency with the labels constructed using SentenceBERT. Secondly, we randomly select 100 samples with labels constructed using SentenceBERT and manually evaluate their consistency. Among them, 92% of the constructed labels are consistent with the human-selected ones.
>
> Furthermore, we conduct experiments using ELECTRA-base to explore the results of labeling with different versions of SentenceBERT. Specifically, we use all-mpnet-base-v2 (420 MB, Avg. Performance: 63.30, SOTA) and all-MiniLM-L6-v2 (80M, Avg. Performance: 58.80, 5 times faster than SOTA). The results are as follows:
> | Method                   | Micro Acc. | Macro Acc. |
> |--------------------------|------------|------------|
> | all-mpnet-base-v2 | 76.0       | 79.9       |
> | all-MiniLM-L6-v2 | 76.2      | 80.5      |
>
> In addition to using the hard labels obtained by selecting the document hypothesis with the highest cosine distance as labeled by SentenceBERT, we also conduct explicit alignment experiments using normalized soft labels and the KL divergence loss function. The results are as follows:
> | Method                   | Micro Acc. | Macro Acc. |
> |--------------------------|------------|------------|
> | Hard Label + Cross Entropy | 76.2       | 80.5       |
> | Soft Label + KL Divergence           | 70.3      | 74.9      |
> In future work, we will continue to explore better alignment methods.
>
> # Question B.1:
> Although the embeddings of document rules are independently used for explicit alignment and many-to-many entailment, the rules can attend to each other as context during encoding. In addition, the attention mechanism is also used to synthesize the information for the final decision classification. Moreover, previous baselines (such as DGM and DISCERN) can be seen as dependently used, but our results are comparable to their best results, and even our main metric is slightly better than theirs.
>
> # Question B.2:
> Thank you for your question. Based on our understanding of decontextualization in the original paper, decontextualization refers to the process of adding crucial contextual information (such as referential and bridging information) to stand-alone sentences, enabling them to maintain their meaning even when detached from context. Its purpose is "to be able to interpret, or present, text independently from the rich context in which it occurs."
>
> In the current CMR task, during the encoding phase, we do not reduce the context for each document fragment (EDU). Each EDU can still see all the surrounding context. The only difference is that after encoding, we use embeddings separately for explicit alignment and many-to-many entailment. In other words, the independent encoding of EDUs that we use still contains contextual information.
>
> However, your question has also prompted us to take decontextualization into consideration if we conduct future experiments on longer documents. If there are limitations on input length and thus we have to split the document, making it impossible to input the complete context, decontextualization may indeed be necessary in such scenarios.
>
>
> # Typos Grammar Style And Presentation Improvements:
> Thank you for your feedback. In the revised version, we will take multiple perspectives into consideration and remove unnecessary description. We will make modifications to simplify the formulations.

---

### Official Review · Reviewer_QzcE · 2023-08-12

**Soundness:** 4

**Excitement:**

4: Strong: This paper deepens the understanding of some phenomenon or lowers the barriers to an existing research direction.

**Paper Topic And Main Contributions:**

This paper presents a two-tiered approach to the Conversational Machine Reading (CMR) task. The first is a Classification (BiAE) model which aligns the document to user-provided data and employs an entailment reasoning module for decision-making. The second utilizes T5 model and BiAE's results for question generation.

Contributions:

1. Introduce an explicit alignment strategy for improved interaction between document content and user data with weakly supervised alignment label.

2. Implement a lightweight many-to-many entailment reasoning module that enhances decision-making based on user queries.

3, A novel approach to generating follow-up questions that consider both the content of the document and previous user questions.

**Questions For The Authors:**

Question A: Did you experiment with varying weights for each term within the joint loss function as mentioned in equation (16)?

Question B: Have you considered an end-to-end strategy that merges the two steps, perhaps through joint training?

Question C: Could you elucidate the rationale behind opting for the maximal cosine distance as the alignment label? (“Specifically, we calculate the cosine distance between sentence vector pairs and select the hypothesis with the maximal cosine distance as the alignment label for each user premise.”)

**Reasons To Accept:**

1. This paper is clear and well structured.

2. It encompasses the latest state-of-the-art baseline models.

3. The introduced two-step methodology offers a fresh perspective, potentially pioneering a new learning paradigm in CMR research.

4. Demonstrable high performance (First place in the leaderboard) on the CMR dataset, ShARC, outperforming existing methods.






**Reasons To Reject:**

The paper does not include experiments on other notable CMR benchmark datasets.

The weakly supervised alignment label, generated using Sentence-BERT, might introduce biases and errors, particularly in longer sentences with negation. It would be beneficial to conduct additional ablation studies, perhaps employing alternative models or introducing noise to this loss, to further investigate its performance."

**Reproducibility:**

4: Could mostly reproduce the results, but there may be some variation because of sample variance or minor variations in their interpretation of the protocol or method.

**Reviewer Confidence:**

4: Quite sure. I tried to check the important points carefully. It's unlikely, though conceivable, that I missed something that should affect my ratings.

---

> ### Author Rebuttal · Authors · 2023-08-29
>
> We are grateful for your thought-provoking comments. We will explain your concerns and answer your questions point by point.
>
> # Reasons To Reject Point 1:
> Thank you for your feedback. Due to limited computational resources, it is challenging for us to organize more experiments. Conducting experiments on additional datasets will be left for our future work. In order to provide more comprehensive evidence of the effectiveness of BiAE, we divide the training set into five parts, conducting a 5-fold cross-validation experiment. We performe statistical significance t-tests between the results of these five experiments and the results of DGM, with p < 0.05. This experiment partially supplements the demonstration of the effectiveness of our approach. Thank you again for your feedback.
>
>
> # Reasons To Reject Point 2:
> Thank you for your feedback. In fact, we have validated the quality of alignment labels through two methods. Firstly, we use a heuristic approach to make labels. Specifically, we calculate the edit distance between all document rules and user information and assign the document rule with the minimum edit distance as the label. The labels generated by the heuristic method show 85% consistency with the labels constructed using SentenceBERT. Secondly, we randomly select 100 samples with labels constructed using SentenceBERT and manually evaluate their consistency. Among them, 92% of the constructed labels are consistent with the human-selected ones. The issue you mentioned may indeed lie within the 8% noise, but this approach has proven to be simple and effective. Thank you again for your feedback.
>
> Furthermore, we conduct experiments using ELECTRA-base to explore the results of labeling with different versions of Sentence-BERT. Specifically, we use all-mpnet-base-v2 (420 MB, Avg. Performance: 63.30, SOTA) and all-MiniLM-L6-v2 (80M, Avg. Performance: 58.80, 5 times faster than SOTA). The results are as follows:
> | Method                   | Micro Acc. | Macro Acc. |
> |--------------------------|------------|------------|
> | all-mpnet-base-v2 | 76.0       | 79.9       |
> | all-MiniLM-L6-v2 | 76.2      | 80.5      |
>
> In addition to using the hard labels obtained by selecting the document hypothesis with the highest cosine distance as labeled by SentenceBERT, we also conduct explicit alignment experiments using normalized soft labels and the KL divergence loss function. The results are as follows:
> | Method                   | Micro Acc. | Macro Acc. |
> |--------------------------|------------|------------|
> | Hard Label + Cross Entropy | 76.2       | 80.5       |
> | Soft Label + KL Divergence           | 70.3      | 74.9      |
> In future work, we will continue to explore better alignment methods.
>
> # Question A:
> Thank you for your question. We conducted several sets of experiments in the preliminary stage to explore the relative importance of different loss components. Specifically, we set up three configurations:
> * (1) L = λ * L_dec + L_align + L_entail,
> * (2) L = L_dec + λ * L_align + L_entail,
> * (3) L = L_dec + L_align + λ * L_entail,
> where λ is a weight parameter. The experimental results showed that L_dec is the most important term. Therefore, we only reported the results of the first configuration (1) in our paper.
>
> # Question B:
> Thank you for your question. We have explored end-to-end modeling by incorporating an reasoning module between the shared encoder and decoder. For the first three types (YES, NO, MORE), the target of the decoder is to generate a special token. We also tried adding adapters for the core decision module and the decoder. The experimental results of using a T5-large encoder, a T5-large decoder, and a BiAE core decision module are as follows.
> | Method                   | Micro Acc. | Macro Acc. | BLEU1 | BLEU4|
> |--------------------------|------------|------------|------------|------------|
> | End2end, Multi-task | 72.56       | 76.96       |40.41       |31.98       |
> | End2end, Multi-task, +Adapter|75.68       | 79.59      | 48.45      | 40.36     |
> The results show that the reasoning task and generation task have mutually negative impacts, and neither can achieve the desired results.
>
> # Question C:
> The cosine distance is commonly used to measure the semantic similarity between sentence pairs. In the vector space of sentence embeddings, vectors that are semantically similar have smaller angles and thus larger cosine distances. The objective of constructing explicit alignment labels is to consider semantic similarity, where we aim to match the most similar document rules with user information. Calculating cosine distances is a simple and effective method, which is why we chose SentenceBERT to obtain sentence embeddings. Additionally, we validated the quality of alignment labels through two approaches. Firstly, we used a heuristic method for labeling, where we calculated the edit distance between all document rules and user information and selected the document with the minimum edit distance as the label. The alignment labels generated by the heuristic method show 85% consistency with the labels constructed using SentenceBERT. Secondly, we randomly selected 100 samples with labels constructed using SentenceBERT and manually evaluated their consistency. Among them, 92% of the constructed labels are consistent with the human-selected ones.

---

### Official Review · Reviewer_AKtE · 2023-08-15

**Soundness:** 4

**Excitement:**

3: Ambivalent: It has merits (e.g., it reports state-of-the-art results, the idea is nice), but there are key weaknesses (e.g., it describes incremental work), and it can significantly benefit from another round of revision. However, I won't object to accepting it if my co-reviewers champion it.

**Missing References:**

I'm not much aware of this area.

**Paper Topic And Main Contributions:**

The paper proposes a an approach in the three stage framework for conversational machine reading. The first stage involves aligning the document context with the user information explicitly, the second stage involves predicting agent action (yes, no, irrelevant or more), and the third stage is generating questions if the agent decides to ask follow up questions.

The authors call the lightweight decision making approach - Bipartite Alignment and many-to-many Entailment.

The method proposed by the authors is lightweight as compared to the baselines and outperforms larger baselines on the ShARC task.

**Questions For The Authors:**

* Did you explore other ways to construct l^{align} instead of simple sentence similarity? Maybe an NLI model?

**Reasons To Accept:**

* The paper is well written and is easy to read and follow
* The method proposed is relatively light-weight and easy to implement. It shows greater performance as compared to more complex baselines.
* The methodology is simple and well described
* The ablations are thorough
* The analysis towards the end is much appreciated and insightful.

**Reasons To Reject:**

* It is unclear if the advantages are from the modeling or data augmentation. Since the central contribution of the paper is in the modeling aspect (explicit alignment), I expect the authors to compare with the baselines where all the baselines have data augmentation for a more fair comparison. I do however note that DA might be a central aspect of this work.
* This work feels like a combination of existing works and not very novel. Since most components (except the explicit alignment) are taken from prior work, I expect the authors to properly and fairly compare other approaches that have the same rest of the components.
* The main results in table 1 don't seem statistically significant. Can you provide statistical tests?

**Reproducibility:**

4: Could mostly reproduce the results, but there may be some variation because of sample variance or minor variations in their interpretation of the protocol or method.

**Reviewer Confidence:**

2: Willing to defend my evaluation, but it is fairly likely that I missed some details, didn't understand some central points, or can't be sure about the novelty of the work.

**Typos Grammar Style And Presentation Improvements:**

line 118 - add and
line 122 - more challenging than?

---

> ### Author Rebuttal · Authors · 2023-08-29
>
> Thanks for your careful and valuable comments. We will explain your concerns and answer your question point by point.
>
> # Reasons To Reject Point 1
> Our pipeline framework consists of a reasoning model and a generation model. DA is only used for training the generation model. The explicit alignment and many-to-many entailment approaches we proposed are designed for the reasoning model. Due to the greater importance placed on the performance of the reasoning model, we have focused extensively on conducting experiments related to the reasoning model. The generation model is only activated when the reasoning result is "MORE." The experiments related to generation account for only a small fraction of all our experiments and are not our primary focus.
>
> # Reasons To Reject Point 2
> The motivation behind our work can be summarized into two points. Firstly, existing methods for reasoning have not taken into account the explicit alignment between document rules and user information, leading to the introduction of noise and sparse attention. Secondly, the core decision modules of existing methods (e.g., DGM using graph networks + transformers) have become increasingly complex, consuming more computational resources and memory. Therefore, in addition to the explicit alignment modeling you mentioned, our main contributions include designing a many-to-many entailment analysis module, significantly reducing the parameter size of the core decision module (only 0.06% of DGM). Experimental results show that our lightweight core decision module achieves comparable results to DGM.
>
> Furthermore, we also conducted ablation experiments on the encoder of the reasoning model, as we recognize the importance of fair comparison. The results of the ablation experiments based on ELECTRA on the development set are as follows. The reasoning results based on ELECTRA are slightly lower than those based on DeBERTa but still higher than DGM. This allows for a fair comparison with the previous SOTA method, DGM, and also demonstrates that the improvement in reasoning performance mainly comes from our incorporation of explicit alignment and the modeling of many-to-many entailment in the core decision module. We will include these experimental results in the revised version. Thank you for your feedback.
> | Model               | Micro Acc. | Macro Acc. |
> |---------------------|------------|------------|
> | DGM (electra-base)  | 75.5       | 79.6       |
> | BiAE (electra-base) | 76.2       | 80.3       |
> | BiAE (DeBERTa-base) | 76.2       | 80.5       |
> | DGM (electra-large) | 78.6       | 82.2       |
> | BiAE (electra-large)| 79.9       | 82.8       |
> | BiAE (deBERTa-large)| 80.5       | 83.2       |
>
>
> # Reasons To Reject Point 3
> Thank you for pointing out the shortcomings. To address this, we have conducted significance test experiments. However, as the test set is reserved (evaluation on the test set requires submitting the model and code to the official ShARC platform), we are unable to perform significance test experiments on the test set. Instead, we divide the training set into 5 folds and conduct a t-test using 5-fold cross-validation, resulting in p < 0.05. We will incorporate these ablation experiments into the revised version.
>
> # Question
> Thank you for your question. As using sentence embeddings to calculate cosine distance is simple and effective, we did not consider other approaches. Exploring better alignment methods will be left for our future work. However, we have conducted quality checks on the alignment labels in two ways.
>
> Firstly, we use a heuristic approach to label the alignments. Specifically, we calculate the edit distance between all document rules and user information and select the document hypothesis with the minimum edit distance as the label. The labels generated by the heuristic method show 85% consistency with the labels generated using SentenceBERT.
>
> Secondly, we randomly select 100 samples where labels are generated using SentenceBERT and manually evaluate their consistency. Among them, 92% of the generated labels are consistent with the manually selected ones.
>
> Furthermore, we conduct experiments using ELECTRA-base to explore the results of labeling with different versions of Sentence-BERT. Specifically, we use all-mpnet-base-v2 (420 MB, Avg. Performance: 63.30, SOTA) and all-MiniLM-L6-v2 (80M, Avg. Performance: 58.80, 5 times faster than SOTA). The results are as follows:
> | Method                   | Micro Acc. | Macro Acc. |
> |--------------------------|------------|------------|
> | all-mpnet-base-v2 | 76.0       | 79.9       |
> | all-MiniLM-L6-v2 | 76.2      | 80.5      |
>
> In addition to using the hard labels obtained by selecting the document hypothesis with the highest cosine distance as labeled by SentenceBERT, we also conduct explicit alignment experiments using normalized soft labels and the KL divergence loss function. The results are as follows:
> | Method                   | Micro Acc. | Macro Acc. |
> |--------------------------|------------|------------|
> | Hard Label + Cross Entropy | 76.2       | 80.5       |
> | Soft Label + KL Divergence           | 70.3      | 74.9      |
> In future work, we will continue to explore better alignment methods.

---

### Official Review · Reviewer_xtUY · 2023-08-16

**Typos Grammar Style And Presentation Improvements:** We utilizes (lines 579-580)
**Soundness:** 3

**Excitement:**

3: Ambivalent: It has merits (e.g., it reports state-of-the-art results, the idea is nice), but there are key weaknesses (e.g., it describes incremental work), and it can significantly benefit from another round of revision. However, I won't object to accepting it if my co-reviewers champion it.

**Paper Topic And Main Contributions:**

The authors propose a pipeline framework that aligns the document with the user-provided information. This framework uses an entailment reasoning module and it can generate follow-up questions based on the document and the previously asked questions. The proposed approach shows the best performance in terms of micro-accuracy on the ShARC dataset.

**Questions For The Authors:**

 Question A: What exactly is meant by the user-provided information and user scenarios (168-169)? It seems that the sample dialogue in Figure 1 is not enough to introduce the concept.
 Question B: How do you define the difference between the heavyweight and lightweight decision modules? You mention that there is a "significant reduction in decision module parameters" (lines 108-109), but how big is the difference?
 Question C: Why was DeBERTaV3 chosen for the experiments and how does it compare to the models used in the baseline methods?
 Question D: How do you generate the direct responses in case of IRRELEVANT, YES, NO labels (lines 175-177)?
 Question E: What is the meaning of "the weakly supervised alignment label" (lines 248-149)? How is it computed?

**Reasons To Accept:**

1. An interesting method that explicitly aligns documents with the user-provided information.
2. It shows SOTA results in terms of micro-accuracy on the ShARC dataset.
3. The paper provides a very good and comprehensive overview of the related work.
4. The authors show the ablation studies and provide a detailed analysis of their results.

**Reasons To Reject:**

1. This paper includes some concepts that are not clearly defined/explained, e.g., the following ones (also see the questions to the authors):
    - User-provided information and user scenarios need to be explained in more detail.
    - The meaning of "the weakly supervised alignment label" (lines 248-149) is not clear.
    - Data augmentation is not well explained. Although the authors claim that "data augmentation improves all generation metrics" (474-475) it's not clear how exactly they augment the data.
2. The statements about the performance improvements seem a little bit over-emphasized. The authors claim that "BiAE significantly outperforms baselines with lightweight decision modules by at least 12.7% in micro accuracy" (090-092). However, on the ShARC leaderboard the difference between the proposed model and the previous SOTA is only 0.5% not 12.7%.
3. The overall computational budget is only partially reported (e.g., no information about how many hours it takes to tune all the parts of the pipeline and the additional costs of the fine-tuning and data augmentation).
4. The authors mention "significant improvements" but as far as I understand they did not perform any significance tests.
5. The paper is slightly longer than 8 pages.

**Reproducibility:**

3: Could reproduce the results with some difficulty. The settings of parameters are underspecified or subjectively determined; the training/evaluation data are not widely available.

**Reviewer Confidence:**

4: Quite sure. I tried to check the important points carefully. It's unlikely, though conceivable, that I missed something that should affect my ratings.

---

> ### Author Rebuttal · Authors · 2023-08-29
>
> We are grateful for your thought-provoking comments. We will explain your concerns and answer your questions point by point.
>
> # Question A & Reasons To Reject Point 1 Bullet point 1 :
> In lines 41-44, we explained that user-provided information includes user scenarios and conversation history. In this regard, we assumed that no additional explanation is needed for user scenarios as the statement itself should be sufficient for understanding. However, your concern is also valid and worth careful consideration. We will address this part specifically in the revised version and provide further explanations.
>
> # Question B:
> We differentiate between heavyweight and lightweight based on the parameter size and structural complexity of the core decision module. From a structural perspective, DGM utilizes a combination of graph networks and Transformers, ET5 uses the T5 Decoder, and DISCERN employs Transformers. In comparison, we only utilize a few linear layers. In terms of parameter size, as mentioned in lines 420-421, our decision module has a parameter size of 31.7K, which is only 0.06% of DGM's parameter size(54M).
>
> # Question C:
> Thank you very much for your question. We chose DeBERTaV3 because of its superior performance on tasks such as RTE. We recognize the importance of fair comparisons, so for the reasoning model, we conducted ablation experiments using an encoder based on ELECTRA on the development set (results below). The experimental results show that the reasoning results based on ELECTRA are slightly lower than those based on DeBERTa but still higher than DGM. This approach allows us to make fair comparisons with the previous SOTA model, DGM, while also demonstrating that the improvement in reasoning performance primarily comes from introducing explicit alignment and the modeling of many-to-many entailment in our core decision module. We will include these ablation experiment results in the revised version. Thank you again for your question.
> | Model               | Micro Acc. | Macro Acc. |
> |---------------------|------------|------------|
> | DGM (electra-base)  | 75.5       | 79.6       |
> | BiAE (electra-base) | 76.2       | 80.3       |
> | BiAE (DeBERTa-base) | 76.2       | 80.5       |
> | DGM (electra-large) | 78.6       | 82.2       |
> | BiAE (electra-large)| 79.9       | 82.8       |
> | BiAE (deBERTa-large)| 80.5       | 83.2       |
>
>
>
> # Question D:
> If the decision result is IRRELEVANT, YES or NO, we will directly reply with the corresponding label itself: "Irrelevant", "Yes", or "No".
>
>
> # Question E & Reasons To Reject Point 1 Bullet point 2 :
> Thank you for your question. "The weakly supervised alignment labels" refer to the labels used for training explicit alignment, which are not provided in the dataset and are constructed using SentenceBERT. Specifically, we compute the cosine distance between all (document rule, user provided information) pairs. For each piece of user provided information, we select the document rule with the maximum cosine distance as its alignment label. Since the labels constructed using this method contain noise (the document rule with the maximum cosine distance may not necessarily be the one that truly needs alignment), they are referred to as "the weakly supervised alignment labels". Additionally, we validate the quality of the weakly supervised alignment labels in two ways. First, we use a heuristic approach to make such labels. Specifically, we calculate the edit distance between all (document rule, user provided information) pairs and assign the document rule with the minimum edit distance as the label for each piece of user provided information. The labels constructed using the heuristic approach show 85% consistency with the labels constructed using SentenceBERT. Second, we randomly selecte 100 samples with labels constructed using SentenceBERT and manually evaluate their consistency. Among them, 92% of the constructed labels match the manually selected ones.
>
> In addition to using SentenceBERT to assign hard labels based on the highest cosine distance, we also conducted an explicit alignment experiment using normalized soft labels and the KL divergence loss function. The results are as follows:
> | Method                   | Micro Acc. | Macro Acc. |
> |--------------------------|------------|------------|
> | Hard Label + Cross Entropy | 76.2       | 80.5       |
> | Soft Label + KL Divergence           | 70.3      | 74.9      |
> In future work, we will continue to explore better alignment methods.
>
> # Reasons To Reject Point 1 Bullet point 3 :
> The method we proposed consists of a pipeline framework with a reasoning model and a generation model. Data augmentation is only used during the training of the generation model. We briefly mentioned data augmentation in lines 338-342, but we may not have explained it clearly enough. Here, we would like to provide a more detailed explanation.
>
> Firstly, the reasoning model produces four types of decision states: YES, NO, IRRELEVANT, and MORE. The generation model is only activated when the decision result is MORE, and only the data with true decision labels as MORE are used for generation. This subset of data accounts for only 31.08% of the training set.
>
> During the generation phase, in order to utilize more training data, we perform data augmentation. Specifically, for the data with true labels of IRRELEVANT, YES or NO, assuming their dialogue history consists of n turns, denoted as H = [(q1, a1), ..., (qn, an)], we shorten the dialogue history by one turn to obtain a new dialogue history, H' = [(q1, a1), ..., (qn-1, an-1)]. We then take the question qn from the last turn as the new ground truth generation question. This helps increase the amount of data available for training the generation model.
>
> # Reasons To Reject Point 2 :
> The previous SOTA method, DGM, used a core decision module consisting of graph networks and Transformers. In contrast, our approach only utilizes a few linear layers, resulting in a core decision module with parameters amounting to only 0.06% of those in DGM. We would like to emphasize that, compared to models using heavyweight decision modules such as DGM, Discern, and ET5, we achieve comparable results. Furthermore, compared to models employing lightweight core decision modules like EMT and E3, we observe a significant improvement of 12.7%.
>
>
> # Reasons To Reject Point 3 :
> Thank you for pointing out the problem. We have calculated the time required for the pipeline of preprocessing, reasoning, and question generation on the Dev set (2270 samples) during the inference stage (using DeBERTa-large+T5-large). The entire inference process takes less than 5 minutes. Preprocessing the data takes about 2 minutes, making the decision classification on the Dev set takes 37 seconds, and generating questions for the data with reasoning results as MORE takes 70 seconds. Except for the preprocessing stage, which runs on the CPU, the GPU configuration and hyperparameters for inference and generation are consistent with what we described in section 4.3.
>
> Additionally, we have calculated the time required for fine-tuning the reasoning model (BiAE) and the generation model. Fine-tuning BiAE (DeBERTa-large) for 5 epochs takes 6-7 hours, fine-tuning T5-large (without data augmentation) for 3 epochs takes 1-2 hours, and fine-tuning T5-large (with data augmentation) for 3 epochs takes 7-8 hours. The data augmentation is performed during the construction of the Dataset and the time required for it is negligible. The GPU configuration and hyperparameters for fine-tuning and reasoning/generation are consistent with what we described in section 4.3.
>
> We will include this information in the revised version. Thank you again for your feedback.
>
>
> # Reasons To Reject Point 4 :
> Thank you for highlighting the shortcomings. To address this, we have conducted significance test experiments. However, since the test set is reserved for official evaluation by ShARC (which requires submission of the model and code), we are unable to perform significance test experiments on the test set. Instead, we divided the training set into 5 folds to conduct 5-fold cross-validation t-tests, with result of p < 0.05. We will include this ablation experiment in the revised version. Thank you again for your feedback.
>
>
> # Reasons To Reject Point 5 :
> Thank you for your attention to detail. According to the submission instructions of EMNLP, the content excluding the Limitations and Ethics Statement sections needs to be kept within 8 pages. We have ensured that our submission complies with this requirement. If it does not meet the specified page limit, our submission may face desk rejection.
>
> Thank you for your meticulousness in identifying typos, grammar, style, and presentation improvements.

---

### Official Review · Reviewer_DMCN · 2023-08-17

**Soundness:** 3

**Excitement:**

2: Mediocre: This paper makes marginal contributions (vs non-contemporaneous work), so I would rather not see it in the conference.

**Paper Topic And Main Contributions:**

The paper proposes a pipeline framework for Conversational Machine Reading (CMR) that addresses the issue of alignment between the document and user-provided information. The proposed method achieves state-of-the-art results on the CMR benchmark dataset ShARC.

**Questions For The Authors:**

+ In comparison with large language models, No examples or demonstrations are employed in the template as shown in Figure 5. Will there be a surge of performance if several shots of examples are given?
+ Why the document is the set of hypotheses while the dialogue history is treated as a premise?
+ The formula and the notation in the manuscript are a little bit cluttered. I suggest deferring the implementation details to the appendix.

**Reasons To Accept:**

+ The paper introduces a novel framework for CMR that explicitly aligns the document and user-provided information, which greatly improves decision-making and question generation.
+ The use of many-to-many entailment reasoning and lightweight decision modules in the pipeline framework is innovative and practical.
+ The experimental results show that the proposed method outperforms baselines and achieves state-of-the-art performance on the ShARC dataset.

**Reasons To Reject:**

+ The potential unfair comparison between the proposed model and the baselines. The proposed method uses a deBERTa for encoding and finetunes a T5 for question generation, which are nearly state-of-the-art backbone models. However, the baseline methods use much weaker backbones, rendering the comparison meaningless.
+ For evaluating question generation, I doubt that simple BLEU metric may be not enough. Human evaluation or at least case study is necessary.

**Reproducibility:**

3: Could reproduce the results with some difficulty. The settings of parameters are underspecified or subjectively determined; the training/evaluation data are not widely available.

**Reviewer Confidence:**

3: Pretty sure, but there's a chance I missed something. Although I have a good feel for this area in general, I did not carefully check the paper's details, e.g., the math, experimental design, or novelty.

---

> ### Author Rebuttal · Authors · 2023-08-29
>
> Thanks for your careful and valuable comments. We will explain your concerns and answer your questions point by point.
>
> # Reasons To Reject Point 1:
> Thank you very much for your thoughtful suggestions. We value fair comparisons a lot. Therefore, for the reasoning model (BiAE), we conducted ablation experiments based on ELECTRA as the encoder on the development set (results shown below). The experimental results indicate that the reasoning (decision) results based on ELECTRA are slightly lower than those based on DeBERTa, but still higher than DGM. This not only allows us to make a fair comparison with the previous SOTA method (DGM), but also demonstrates that the improvement in reasoning performance mainly comes from the introduction of explicit alignment and modeling of many-to-many entailment in our core decision module. It should be noted that the parameter size of our core decision module (31.7K) is only 0.06% of DGM (54M). One of our main contributions lies in significantly reducing the parameter size of the core decision module while achieving comparable results to DGM. Additionally, we have conducted significance test using 5-fold cross-validation with the result of p < 0.05 in a t-test. We will incorporate these ablation experiments into the revised version. Once again, thank you for your feedback.
> | Model               | Micro Acc. | Macro Acc. |
> |---------------------|------------|------------|
> | DGM (electra-base)  | 75.5       | 79.6       |
> | BiAE (electra-base) | 76.2       | 80.3       |
> | BiAE (DeBERTa-base) | 76.2       | 80.5       |
> | DGM (electra-large) | 78.6       | 82.2       |
> | BiAE (electra-large)| 79.9       | 82.8       |
> | BiAE (deBERTa-large)| 80.5       | 83.2       |
>
>
> # Reasons To Reject Point 2:
> We conducted a case study on 100 samples with the lowest BLEU scores to analyze the reasons, and the main categories are summarized as follows:
> * (1)	Incomplete generation of questions: 2%
> * (2) Generated questions lacking key words: 2%
> * (3) Generated questions lacking non-key words: 8%
> * (4) Generated questions with the same semantics as the true questions but different expressions: 8%
> * (5) Generated questions unrelated to the document: 13%
> * (6) Redundant generation of questions (already asked or unnecessary): 25%
> * (7) Generated questions describing other reasonable and unasked conditions: 42%
>
> As you have rightly pointed out, evaluating the generation task solely based on BLEU is not sufficient. Categories 4 and 7 (50% in total) represent acceptable generated questions but receive lower BLEU scores. We will explain this issue adequately in the revised version. Thank you for your thoughtful considerations.
>
> # Question 1:
> Thank you for your question. The purpose of evaluating on the large language models was to assess its ability to solve the CMR task. Since the best results on the large language models (GPT-3.5, Micro-Acc 35.6 Macro-Acc 45.2) were still far below our results (Micro-Acc 80.5, Macro-Acc 83.2), we did not conduct additional few-shot experiments before submitting the paper. However, your question has motivated us to perform few-shot experiments and supplement the results for the large language models, which will make our findings more rigorous.
>
> The results of the 2-shot experiments on the large language models are as follows. The results show that providing examples can improve the performance, but the best results are still far below ours.
> | LLM              | Micro Acc. | Macro Acc. | BLEU-1 | BLEU-4 |
> |--------------------|------------|------------|--------|--------|
> | FastChat-T5 (3B)   | 12.7       | 29.8       | -      | -      |
> | + 2-shot           | 20.6       | 35.9       | 50.9   | 29.5   |
> | Alpaca (7B)        | 51.8       | 37.7       | -      | -      |
> | + 2-shot           | 41.8       | 30.2       | -      | -      |
> | Vicuna (13B)       | 24.8       | 39.0       | 11.9   | 7.7    |
> | + 2-shot           | 33.1       | 41.9       | 33.8   | 12.0   |
> | GPT-3.5 (175B)     | 35.6       | 45.2       | 58.6   | 37.1   |
> | + 2-shot           | 41.1       | 50.8       | 45.2   | 29.5   |
>
>
> # Question 2:
> We have excerpted the definition proposed by previous researchers: "Textual entailment is defined as a directional relationship between pairs of text expressions, denoted by T (the entailing 'Text') and H (the entailed 'Hypothesis'). We say that T entails H if humans reading T would typically infer that H is most likely true". (Please refer to Ido Dagan; Dan Roth; Fabio Zanzotto; Mark Sammons, Morgan & Claypool, 2013. Recognizing Textual Entailment: Models and Applications) A hypothesis is a statement that can be tested and verified through empirical evidence, while a premise is a statement used to support or justify a conclusion in an argument. In the CMR task, evidence is continuously provided through dialogue to determine if the user-provided information can entail the document. For the sake of convenience, we consider the document as a set of hypotheses to be verified, and the information provided by the user as premises to support the verification of the document.
>
> # Question 3:
> Thank you for your feedback. We will carefully review all the formulas and expressions again, and strive to present them in a more concise manner whenever possible. Regarding the suggestion of placing implementation details in the appendix, we have also considered this before submitting the paper. Once again, thank you for your suggestions, and we will reorganize the placement of this section in the revised version.

---

### Meta-Review · Area_Chair_woN7 · 2023-09-19

**Recommendation:** 3

**Metareview:**

Based on the reviews, the paper proposes a pipeline framework for Conversational Machine Reading (CMR) that addresses the issue of alignment between the document and user-provided information. The proposed method achieves state-of-the-art results on the CMR benchmark dataset ShARC.

The reviewers generally appreciate the proposed framework, highlighting its novelty, practicality, and improved performance over baselines. They also commend the clear writing style and thoroughness of the experiments and analysis.

However, there are some concerns raised by the reviewers. One reviewer questions the potentially unfair comparison between the proposed model and baselines due to the differences in backbone models. Another reviewer raises questions about the clarity of certain concepts and urges the authors to provide more details and clarity. Furthermore, there are suggestions to include more datasets for evaluation and to consider alternative models or noise in the weakly supervised alignment label.

In terms of novelty, some reviewers express ambivalence, noting that while the proposed many-to-many entailment reasoning module is novel, other components of the pipeline are incremental. The pros mentioned by the reviewers include the clear writing, improved performance, and insights provided by the ablation studies.

In summary, the paper presents a promising pipeline framework for CMR, addressing the alignment issue and achieving state-of-the-art results on the ShARC dataset. The reviewers appreciate the clear writing and thorough experiments, but some concerns and suggestions for improvement have been raised.

---

### Decision · Program_Chairs · 2023-10-07

**Decision:**

Accept-Findings

**Comment:**

Based on the reviews, the paper proposes a pipeline framework for Conversational Machine Reading (CMR) that addresses the issue of alignment between the document and user-provided information. The proposed method achieves state-of-the-art results on the CMR benchmark dataset ShARC.

The reviewers generally appreciate the proposed framework, highlighting its novelty, practicality, and improved performance over baselines. They also commend the clear writing style and thoroughness of the experiments and analysis.

However, there are some concerns raised by the reviewers. One reviewer questions the potentially unfair comparison between the proposed model and baselines due to the differences in backbone models. Another reviewer raises questions about the clarity of certain concepts and urges the authors to provide more details and clarity. Furthermore, there are suggestions to include more datasets for evaluation and to consider alternative models or noise in the weakly supervised alignment label.

In terms of novelty, some reviewers express ambivalence, noting that while the proposed many-to-many entailment reasoning module is novel, other components of the pipeline are incremental. The pros mentioned by the reviewers include the clear writing, improved performance, and insights provided by the ablation studies.

In summary, the paper presents a promising pipeline framework for CMR, addressing the alignment issue and achieving state-of-the-art results on the ShARC dataset. The reviewers appreciate the clear writing and thorough experiments, but some concerns and suggestions for improvement have been raised.